# Compact Wideband Four-Port MIMO Antenna for Sub-6 GHz and Internet of Things Applications

**DOI:** 10.3390/mi13122202

**Published:** 2022-12-12

**Authors:** Nathirulla Sheriff, Sharul Kamal, Hassan Tariq Chattha, Tan Kim Geok, Bilal A. Khawaja

**Affiliations:** 1Wireless Communication Center, School of Electrical Engineering, Faculty of Engineering, Universiti Teknologi Malaysia, Johor Bahru 81310, Malaysia; 2Advanced Cyclotron Systems Inc. (ACSI), Richmond, BC V6X 1X5, Canada; 3Faculty of Engineering and Technology, Multimedia University, Melaka 75450, Malaysia; 4Department of Electrical Engineering, Faculty of Engineering, Islamic University of Madinah, P.O. Box 170, Madinah 41411, Saudi Arabia

**Keywords:** 5G, MIMO, IoT devices, Wi-Fi, four-ports, sub-6 GHz

## Abstract

A compact four-port multi-input, multi-output (MIMO) antenna with good isolation is proposed for sub-6 GHz and Internet of Things (IoT) applications. Four similar L-shaped antennae are placed orthogonally at 7.6 mm distance from the corner of the FR4 substrate. The wideband characteristics and the required frequency band are achieved through the L-shaped structure and with proper placement of the slots on the substrate. To obtain good isolation between the ports, rectangular slots are etched in the bottom layer and are interconnected. The proposed antenna has total dimensions of 40 mm × 40 mm × 1.6 mm. The interconnected ground plane provides good isolation of less than −17 dB between the ports, and the impedance bandwidth obtained by the proposed four-port antenna is about 54% between the frequency range of 3.2 GHz to 5.6 GHz, thus providing a wideband antenna characteristic covering sub-6 GHz 5G bands (from 3.4 to 3.6 GHz and 4.8 to 5 GHz) and the WLAN band (5.2 GHz). The proposed design antenna is fabricated and tested. Good experimental results are achieved when compared with the simulation results. As the proposed design is compact and low profile, this antenna could be a suitable candidate for 5G and IoT devices.

## 1. Introduction

With a large number of users and the rapid development of wireless communication technologies, higher data rates and channel capacities are in great demand [1,2]. Multiple antennae integrating in the same portable device is seen as a hopeful solution, which could enhance communication network quality and channel capacity. Hence, multi-input, multi-output (MIMO) technology plays a key role in the 5G research hotspot. The European Commission (EC) announced that the band from 3.4 to 3.8 GHz was allocated for 5G, and similarly the Ministry of Industry and Information Technology of China has also considered 3.3–3.6 GHz and 4.8–5 GHz as the operation frequency bands of the 5G system [3]. Recently, many MIMO antenna designs for 5G sub-6 GHz were reported in the literature [4,5,6,7,8,9,10,11,12], but these antennae provide less bandwidth or higher mutual coupling. Contradictorily, the mutual coupling reduction and low envelope correlation coefficients (ECCs) between nearby antenna elements could increase the antenna size, and hence these factors play a key role in antenna design for portable devices. Hence, embedding multiple antennae inside the device in a limited space while maintaining good isolation becomes an antenna design challenge for portable devices.

Different techniques were presented in [13] to reduce the mutual coupling. In order to enhance the isolation, parasitic elements [14,15] are placed between radiating elements to create extra coupling paths. Defected ground structures [16] inhibit surface waves to reduce mutual coupling between the antenna elements by acting as band-stop filters. However, this technique decreases the total antenna efficiency. The etching of slots [17] disturbs the surface current distribution and the path length, which reduces the electromagnetic energy coupling between the ports. Neutralization lines [18] are employed for isolation enhancement by creating an extra coupling path suitable for narrow band decoupling. In [19,20] high isolation is achieved through the orthogonal polarization diversity technique using different excitation modes, while in [21] the multimode decoupling technique is employed to improve the isolation between the antenna elements. However, these designs work only for a single band or less bandwidth. In [7,9,22,23], an antenna is designed for multiple bands for sub-6 GHz applications. Good isolation is achieved using the slotted ground plane method in [24], and similarly in [25], the rectangular slot is etched in the ground plane to stop the flow of current. Moreover, the antenna designs presented were either complex in structure or larger in size and thus integration into a compact MIMO structure for portable devices could be challenging. Therefore, a unique antenna design with the features of extended bandwidth and good isolation suitable for sub-6 GHz and IoT applications needs to be investigated urgently.

In this paper, a compact four-port wideband MIMO antenna design is presented, with four antenna elements positioned near each other in a symmetric fashion with a common ground plane. Simple techniques of etching the slots are used in the top layer and the ground plane to attain the required impedance bandwidth and enhance the isolation between the ports. A peak gain lies between 2.4 to 4.9 dBi for the entire operational bandwidth and the average radiation efficiency obtained is 93%. ECC achieved is less than 0.05, which satisfies the IEEE standards [6] for MIMO antennae for portable devices. The impedance bandwidth obtained by the proposed four-port antenna is about 54%, which ranges from 3.2 GHz to 5.6 GHz, thus providing wideband antenna characteristics covering sub-6 GHz 5G bands (from 3.4 to 3.6 GHz and 4.8 to 5 GHz) and the WLAN band (5.2 GHz).

## 2. Antenna Design

A compact four-port MIMO antenna is designed and fabricated on a FR4 substrate with thickness (*t*) = 1.6 mm, loss tangent (*tanδ*) = 0.025, and dielectric constant (*ε_r_*) = 4.4. Figure 1 illustrates the geometry of the proposed antenna, and the optimum parameters related to the proposed antenna design are listed in Table 1. A simple decoupling structure is implemented in the ground plane to obtain good isolation. CST Microwave Studio has been used for simulation purposes to design and analyze the antenna parameters. The total dimensions of the MIMO antenna are 40 mm × 40 mm (~0.59λ × 0.59λ at center frequency of 4.45 GHz). The design stages are demonstrated in the subsequent sections.

### 2.1. Single-Port Antenna

An initial configuration of the proposed MIMO system with a single-port antenna is shown in Figure 2C and its total dimensions is 20 mm × 20 mm × 1.6 mm. The design-evolution steps of the proposed antenna are shown in Figure 2. In Figure 3, the return-loss (*S*_11_) results obtained in each evolution step while designing the proposed MIMO antenna are shown.

Initially, an antenna with less than half ground plane and an L-shaped antenna element is designed, as shown in Figure 2A. It can be observed that the impedance bandwidth of 26% is achieved from 4.6 to 6 GHz (*S*_11_ < −10 dB). In the next steps, the ground plane is modified (Figure 2B) and then further improved (Figure 2C) to radiate for the required frequency band. The S-parameter plot in Figure 3 shows that *S*_11_ < −8 dB is achieved for the complete required frequency range from 3.5–5.4 GHz with impedance bandwidth of 43%.

### 2.2. Four-Port MIMO Antenna

The four-port MIMO antenna is proposed from the single-port antenna design discussed in the preceding section. At the initial stage, four antennae are placed orthogonally to each other on the top layer of the FR4 substrate, as shown in Figure 4, and the total volume of the antenna is 40 mm × 40 mm × 1.6 mm. Each antenna element with its feeding port of width (w_port) = 3 mm is placed at a distance of 7.6 mm from the corner end of the substrate. The inter-element spacing between the two antenna elements is 12 mm. The dimensions are properly adjusted in such a way to achieve good bandwidth covering the required frequency range and to obtain good isolation.

The step-by-step configuration of the proposed antenna is shown in Figure 4. The simulated S-parameters plotted in Figure 5A clearly show that the return loss of −10 dB starts only from 4.5 GHz and from 4.2 GHz in steps 1 and 2, respectively. Similarly, both the step designs (STEP 1 and STEP 2) have high mutual coupling between the ports (Figure 5B). In order to achieve good isolation, the ground plane is modified as shown in Figure 4 (steps 2 and 3) by arranging a slot in the center of the antenna ground plane, connected to each other to form a common ground plane. It is also observed that, the currents almost penetrate between the nearby antenna elements in step 2 compared to current distribution in step 3. Good isolation and impedance bandwidth is achieved in the proposed design of step 3 (Figure 5B).

## 3. Results and Discussion

The fabricated four-port MIMO antenna is shown in Figure 6A (top view) and Figure 6B (bottom view). Using an Agilent PNA-X N5242A vector network analyzer (VNA), the S-parameters are measured. The radiation pattern is measured in an anechoic chamber by using a Nanjing Lopu Co. antenna measurement system. The measurement scenario of the proposed antenna is given in Figure 7 to show the measurement environment. The simulated and measured return loss for the designed antenna is represented in Figure 8A. The figure representation clearly shows that the simulated *S*_11_ results of all the four ports are the same due to its similar structure, and good impedance bandwidth of 57% is achieved between the frequency range of 3.2 GHz to 5.8 GHz.

On the other side, measured return-loss results of the proposed antenna show utmost similar results with good bandwidth of 54% covering the frequency range from 3.2 GHz to 5.6 GHz. The measured return-loss results show a slight difference in the frequency range. This difference is primarily due the fabrication process and slight alteration in the dielectric constant of the substrate.

Similarly, the simulation and measured isolation results between the antenna elements are shown in Figure 8B, and isolation between the antenna elements are greater than 16 dB throughout the expected frequency, which demonstrates that all the four antenna elements work independently. A sequence of parameter analyses is presented on the proposed MIMO antenna system to understand the process of the design principle. In Figure 5, the purpose of the rectangular slot at the ground plane is studied with *S*_11_ and *S*_12_ measurements by etching with and without the slot in the ground layer. The use of slot C in the ground not only improves the isolation to −22 dB but also enhances the frequency bandwidth. As slot C in the ground plane increases, the frequency bandwidth of the MIMO antenna gradually increases with the frequency range gradually moving back from 4.3 GHz to 3.3 GHz, and similarly the bandwidth also increases from 0.4 GHz to 2.5 GHz, as shown in Figure 9A. Similarly, to understand the effects of using slot E etched in the ground plane, the dimensions of slot E are adjusted from 2 mm to 7 mm while maintaining all other parameter values unchanged. It can be seen in Figure 9B, that the return-loss *S*_11_ gradually decreases to −10 dB covering the entire frequency range from 3.3 GHz to 5.8 GHz.

Figure 10B shows the working principle of the proposed four-port antenna with the surface current distribution for different frequency bands. This indicates that with the proposed antenna design, the surface current almost does not transfer between the nearby antenna elements at 3.4 GHz, 4.8 GHz, and 5 GHz. This feature assures good isolation between the antenna elements.

Figure 11A–D depicts the simulated and measured 2D YZ-plane and XZ-plane radiation patterns with port 1 excited at 3.4 GHz and 4.8 GHz. The other ports are connected to the 50–ohm match load. It is obvious that the radiation patterns of both the simulated and measured are similar while port 1 is excited and are radiating omnidirectionally. The peak gain achieved by all the four ports lies between 2.4 to 4.9 dBi over the entire operational bandwidth and the average radiation efficiency obtained is 93%. From Figure 11A,B, it can be seen that the maximum measured gain of 2.6 dBi is achieved in the YZ and XZ planes. Similarly, a maximum measured gain of 4 dBi at 4.8 GHz is achieved, as shown in Figure 11C,D.

The ECC and the diversity gain (DG) are important parameters to assess the performance of the MIMO system. The mutual coupling and return loss at the ports can be used to determine ECC, which helps to find the diversity performance of the MIMO antennae [23,24], and is given in Equation (1):(1)$$\left|\rho_{e}\left(i,j,N\right)\right|=\frac{\left|\sum_{n=1}^{N}S_{i,n}^{*}S_{n,j}\right|}{\sqrt{\left|\Pi_{k\left(=i,j\right)}\lceil 1-\sum_{n=1}^{N}S_{i,n}^{*}S_{n,k}\rceil \right|}}$$

The correlation can also be measured from MIMO antenna’s far-field radiation patterns [25], as given in Equation (2):(2)$$\mathrm{ECC}=\frac{{|\int^{\;}\int_{0}^{4}\left[E_{i}\left(\theta ,\phi \right)*E_{j}\left(\theta ,\phi \right)\right]d\Omega |}^{2}}{\int^{\;}\int_{0}^{4}|{E_{i}\left(\theta ,\phi \right)|}^{2}d\Omega \int^{\;}\int_{0}^{4}|{E_{j}\left(\theta ,\phi \right)|}^{2}d\Omega }$$
where *i* and *j* are the antenna elements and *N* is the number of antennae. $E_{i}\left(\theta ,\phi \right)$ and $E_{j}\left(\theta ,\phi \right)$ are the three-dimensional radiation patterns of *i*th and *j*th antenna and Ω is the solid angle. The acceptable and standard value of ECC should be less than 0.5 for portable devices. Similarly, the antenna DG is a well-known performance parameter used to verify the efficacy of the diversity [26]. It can be defined as the ratio of rise in SNR of mixed signals from multiple antennae to the SNR from a single antenna in the system. The DG can be calculated using Equation (3):(3)$$DG=\sqrt[{10}]{1-{\left|ECC\right|}^{2}}\;$$

It can be observed that the ECC is less than 0.005 and DG is greater than 9.9 dB in the 3.4 to 6.5 GHz frequency band, as shown in Figure 12. This signifies good diversity performance and shows good performance results in the achieved frequency band. Table 2 provides a comparison between the proposed wideband MIMO antenna and other antenna designs [27,28,29,30,31,32,33,34] found in the literature. This comparison clearly indicates that the proposed antenna design is exceedingly competitive with other designs discussed in the literature in terms of impedance bandwidth, size, and isolation, along with good values of ECC and diversity gain.

## 4. Conclusions

A four-port fabricated compact MIMO antenna with interconnected ground plane and simple decoupling structure is proposed and developed covering different sub-6 GHz bands, including 5G and Wi-Fi bands. The measured results show that the impedance bandwidth of 54% (3.2–5.6 GHz) and approximate peak gain of 2.4 to 4.9 dBi over the entire operational bandwidth is achieved. Simple decoupling structure provided good, measured isolation results, better than 16 dB, for the proposed four-port MIMO antenna system, even though the antenna elements are placed close to each other. Furthermore, the measured results and the radiation patterns ensure that the fabricated MIMO antenna system provides a good solution for compact sub-6 GHz MIMO portable devices with diversity performance and for IoT devices.

## Figures and Tables

**Figure 1 micromachines-13-02202-f001:**
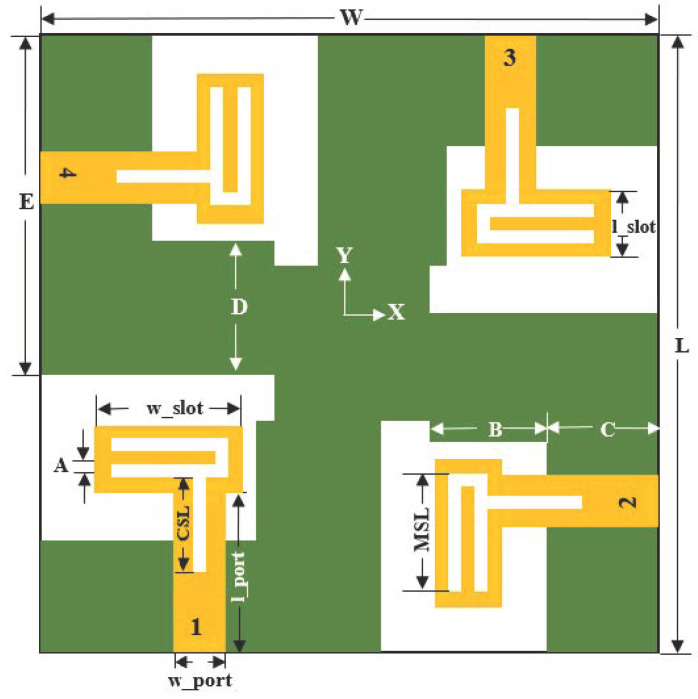
Structure and dimensions of proposed 4 × 4 MIMO antenna.

**Figure 2 micromachines-13-02202-f002:**
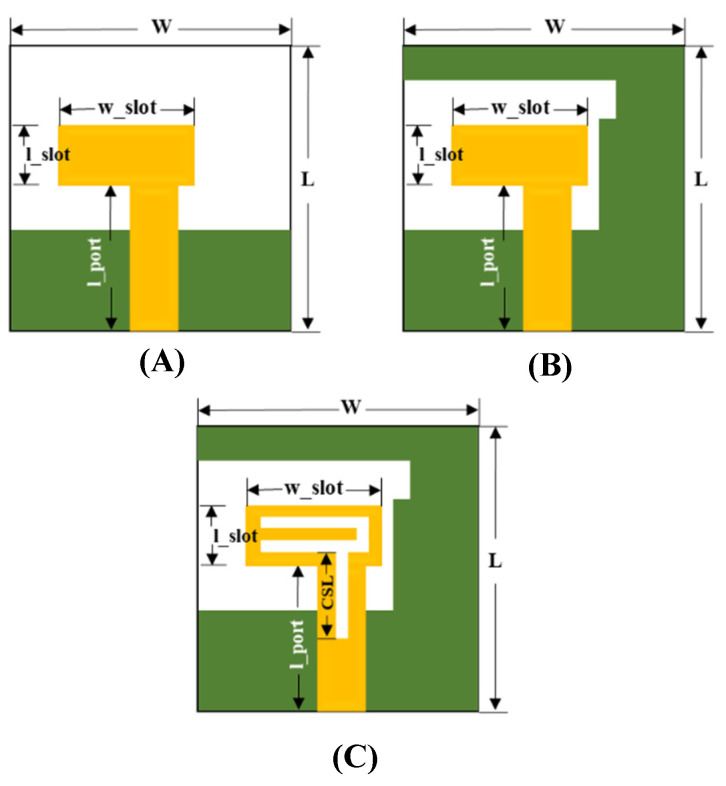
Design-evolution process of the single-element antenna. (**A**) Step–1, (**B**) Step–2, (**C**) Step–3.

**Figure 3 micromachines-13-02202-f003:**
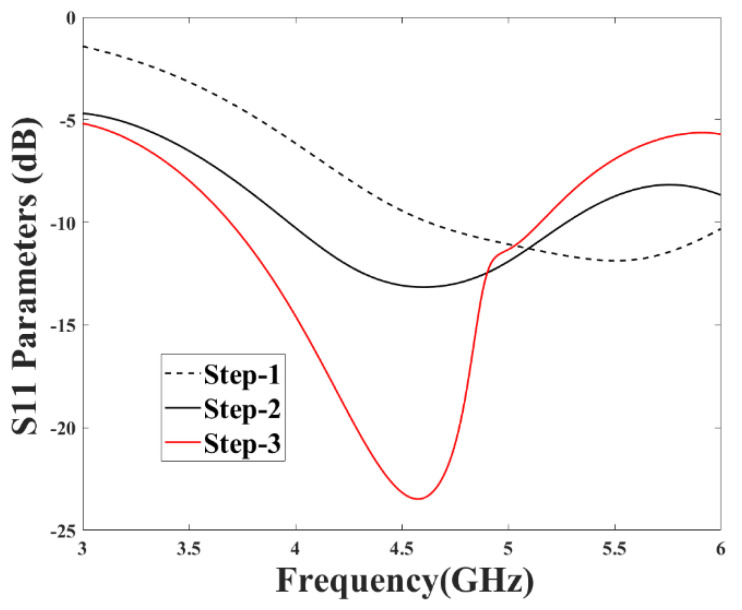
S–parameters for different configuration.

**Figure 4 micromachines-13-02202-f004:**
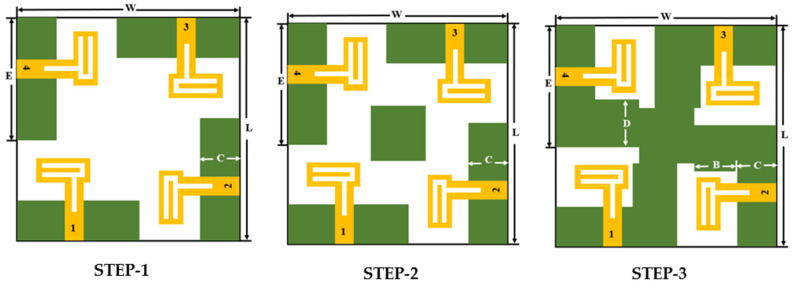
Design evolution of the ground plane structure for the proposed antenna.

**Figure 5 micromachines-13-02202-f005:**
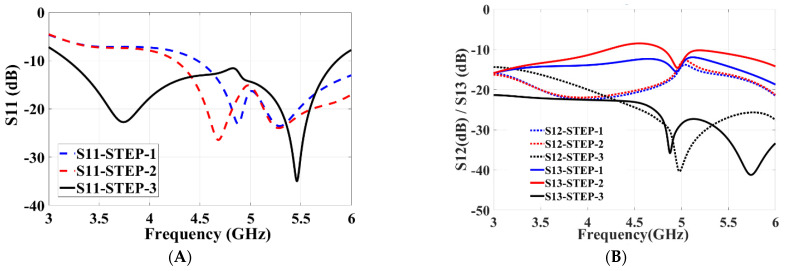
Frequency variation of S-parameters for different configuration: (**A**) *S*_11_ [dB], and (**B**) *S*_13_ [dB].

**Figure 6 micromachines-13-02202-f006:**
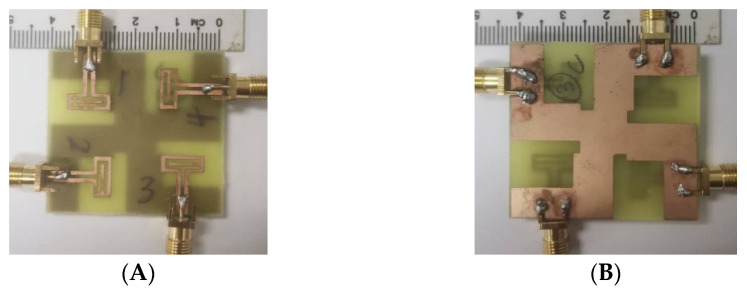
Fabricated prototype of the proposed antenna design: (**A**) top view and (**B**) bottom view.

**Figure 7 micromachines-13-02202-f007:**
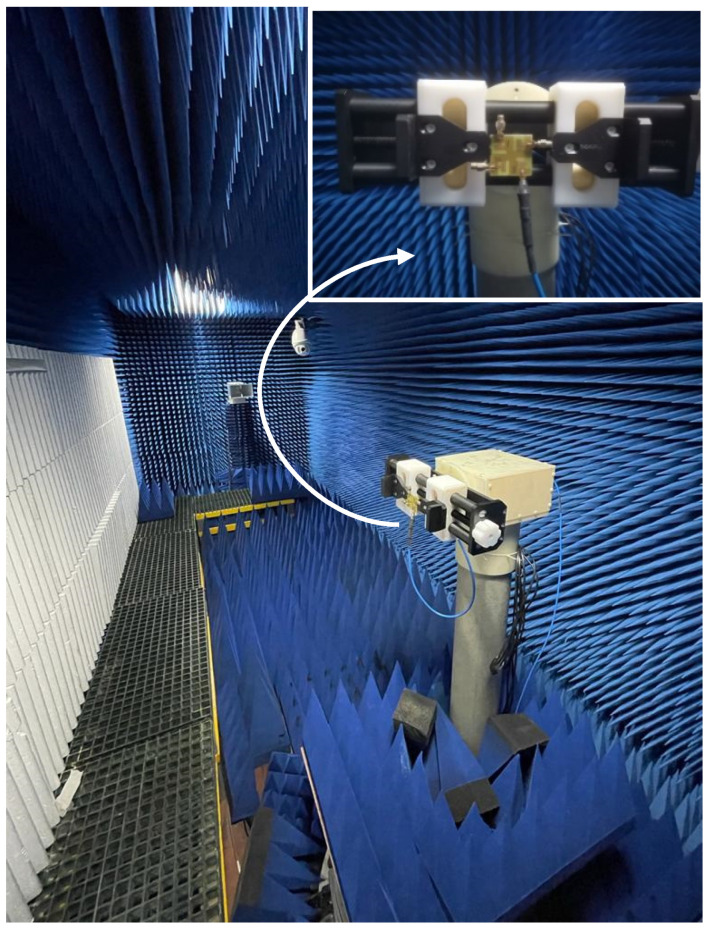
MIMO antenna measurement setup in an anechoic chamber.

**Figure 8 micromachines-13-02202-f008:**
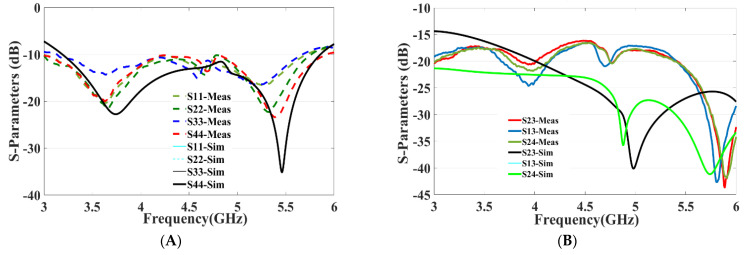
Simulated and measured results of the proposed antenna: (**A**) *S*_11_, *S*_22_, *S*_33_, *S*_44_ and (**B**) *S*_13_, *S*_23_, *S*_23_, *S*_24_.

**Figure 9 micromachines-13-02202-f009:**
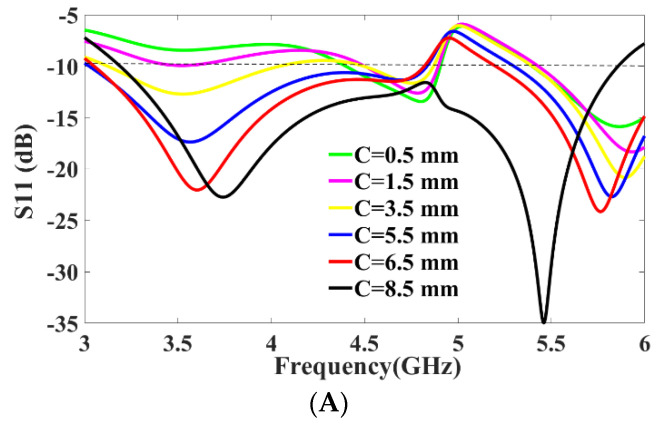
Parametric Analysis: (**A**) *S*_11_ [dB] vs. Frequency [GHz] for slot C parameter values, (**B**) *S*_11_ [dB] vs. Frequency [GHz] for slot E parameter values.

**Figure 10 micromachines-13-02202-f010:**
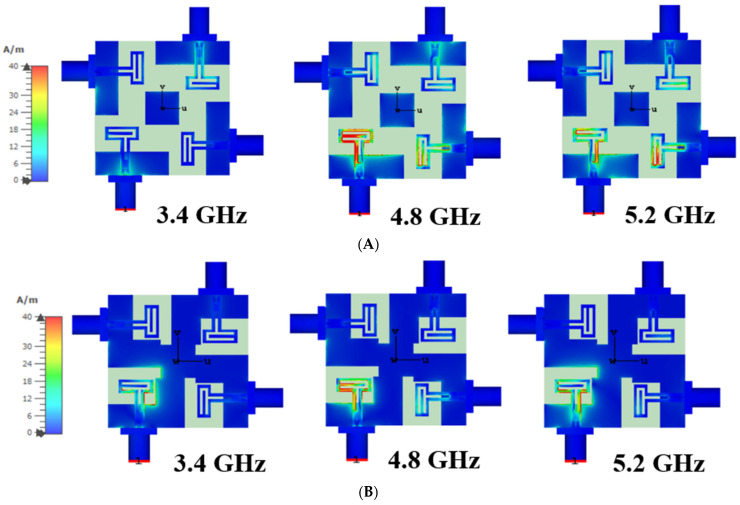
Current distributions in 3.4 GHz, 4.8 GHz and 5.2 GHz. (**A**) Step 2 design antenna; (**B**) final proposed antenna.

**Figure 11 micromachines-13-02202-f011:**
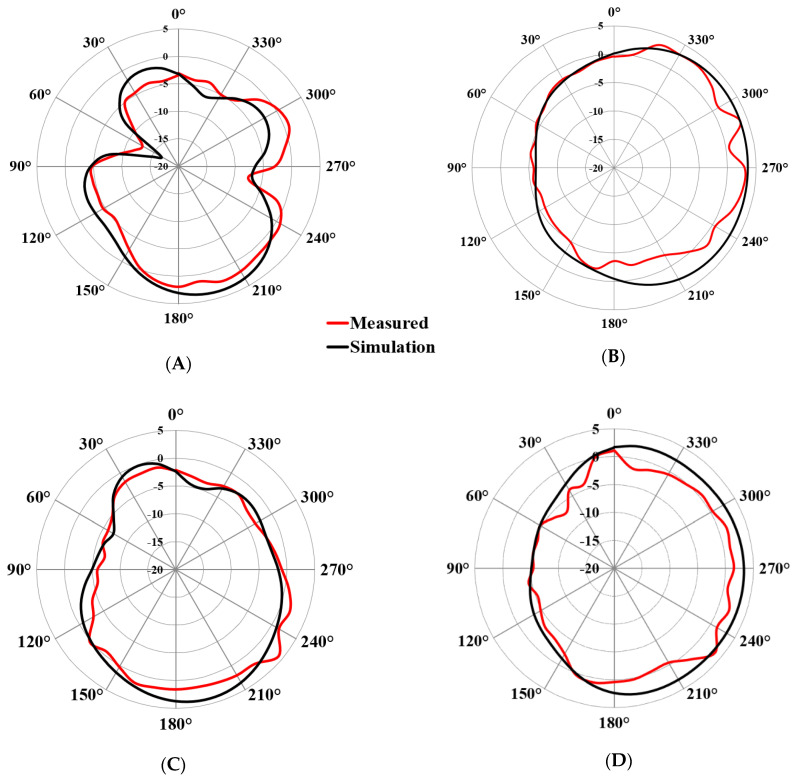
Simulation and measured radiation pattern for port 1: (**A**) 3.4 GHz at YZ plane, (**B**) 3.4 GHz at XZ plane, (**C**) 4.8 GHz at YZ plane, and (**D**) 4.8 GHz at XZ plane.

**Figure 12 micromachines-13-02202-f012:**
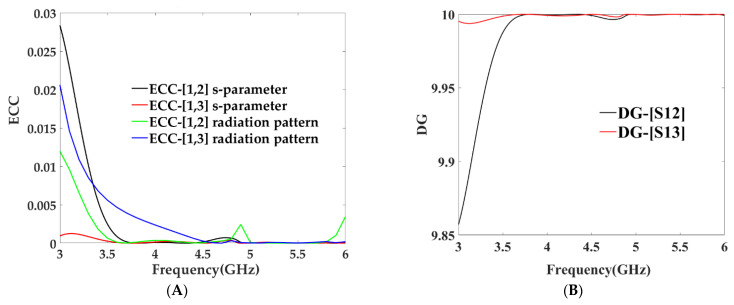
(**A**) Envelope correlation coefficient [dB] vs. frequency [GHz] (**B**) diversity gain [dB] vs. frequency [GHz].

**Table 1 micromachines-13-02202-t001:** Antenna dimensions of the proposed design (mm).

Parameter	Dimension (mm)
*L*	40
*W*	40
*CSL*	6.5
*MSL*	8
*l_port*	10
*w_port*	2.8
*A*	0.8
*B*	8
*C*	7
*D*	8.5
*E*	22
*W_slot*	10
*L_slot*	4

**Table 2 micromachines-13-02202-t002:** Comparison between the proposed antenna and other antenna designs in the literature.

Reference/Year	Isolation (dB)	Bandwidth (GHz)	ECC	Isolation/Diversity Technique	Impedance Bandwidth	Total Antenna Size(mm)	Inter-Element Spacing	Common Ground	Number of Ports
[27]/[2020]	19	4.3–6.5	0.004	Orthogonal placement	40%	50 × 50	~0.18λ	No	4
[28]/[2019]	15	2.4, 5.2 and 5.8	0.5	Orthogonal placement	-	52 × 50	~0.10λ	No	4
[29]/[2019]	15	3–10.74	0.1	Parasitic T-shaped strip	112%	81 × 87	~0.77λ	No	4
[30]/[2019]	17	4.58–6.37	0.05	Parasitic C-shaped	32%	40 × 36	~0.20λ	No	4
[31]/[2019]	15	5.1–5.7	0.05	DGS/decoupling network	11%	50 × 27	~0.16λ	No	4
[32]/[2019]	13	3.3–4.2	0.06	Slots/stubs	24%	42 × 42	~0.15λ	Yes	4
[33]/[2022]	20	3.2–5.7	0.002	EBG	56%	46 × 46	~0.3λ	Yes	4
[34]/[2017]	15	2.3–3.2 and 5.4–5.6	0.05	Polarization diversity/SRR	36%	40 × 40	~0.18λ	Yes	4
This work/[2022]	15	3.2–5.5	0.005	Polarization diversity	54%	40 × 40	~0.17λ	Yes	4

## Data Availability

Not applicable.

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
