# Peer review of "Compact Wideband Four-Port MIMO Antenna for Sub-6 GHz and Internet of Things Applications"

_micromachines, 2022, doi:10.3390/mi13122202_

Round 1

Reviewer 1 Report

1. The ECC should be calculated from measured complex radiation pattern,such as paper “Wideband MIMO antenna with enhanced isolation for wireless communication application,https://doi.org/10.1587/elex.15.20180948”.

2. The cross-polarization is important in MIMO system that should be given.

3. There are many related papers about compact wideband MIMO Antennas, please explain the innovation points, and the advantages of this article.

Reviewer 2 Report

A compact 4-port MIMO antenna was proposed for sub-6 GHz and internet of things (IoT) applications. The antenna was fabricated and the simulation and measured results were compared. However, there are some questions and comments regarding to the manuscript as follows:

1- According to the Figures, it seems that the antenna elements are "L" shaped, not "T" shaped.

2- The authors have claimed that "The S-parameters plot in Figure 3 shows that

S11 < -10 dB is achieved for the complete required frequency range from 3.5–5.7 GHz with impedance bandwidth of 49.5%".  The impedance bandwidth in Fig. 3 is less than the mentioned value.

3- The size of the numbers for the horizontal and vertical axes of Figure 3 is small.

4- In Fig.4, the ground plane of four antennas are connected together for "STEP-3". Therefore, the propagation of surface waves increases, and we expect the isolation decreases. Why the isolation between the ports is increased for "STEP-3", as shown in Fig. 5B?

5- The isolation between the adjacent elements (S12) should also be added to Figure 5.

6- The four MIMO antenna ports are symmetrical; however the difference between the measurement results for S11, S22, and S33 in Fig. 8A is considerable. Please explain the reason.

7- In the paragraph after Fig. 8, slot "C" and slot "E" are not determined. where are these slots?

8- The following sentence in the manuscript is incomplete:

"The working principle of the proposed four antenna MIMO system, the simulated current distributions of the proposed MIMO antenna at 3.4 GHz, 4.8 GHz and 5 GHz are given in Figure 10."

9- The patterns are shown in Fig. 11 in the XZ and YZ planes, but these planes are not determined anywhere.

10. Innovation of the antenna is weak. What is the novelty of the antenna? The techniques used in the paper can be found in previous research.

Reviewer 3 Report

This work presents a four-port wideband MIMO antenna with four symmetrically placed antenna elements next to one another and a common ground plane. The manuscript includes both simulation and measurement data to explain the work. The manuscript has the potential to be accepted for publication. However, updates are required before the resubmission.

First of all, the authors didn’t format the manuscript properly for peer review. The line numbers are missing and an important section is also missing from the first page including the copyright section. Please format the manuscript according to the journal’s template properly for a smooth process from peer review to publication.

The authors claimed in the manuscript that the proposed antenna is a good candidate for the internet of things (IoT) applications. But they didn’t explain how the proposed antenna is a good candidate for IoT applications. Please explain this in the manuscript. And what is the frequency range for the antennas for IoT applications? Add them to the manuscript.

In the introduction section, the authors should add more recently published MIMO antennas for the literature review. Here are some suggestions. Mutual Coupling Reduction of a Circularly Polarized MIMO Antenna Using Parasitic Elements and DGS for V2X Communications, IEEE Access, 2022; MIMO Antenna with Improved Isolation for ISM, Sub-6 GHz, and WLAN Application, Micromachines, 2022; The authors also can find related articles by internet search.

The value of an important parameter of the antenna design (tanδ of the substrate) is missing in the antenna design section. Please provide the value of tanδ of the substrate in the manuscript.

Antenna parameter symbols in both Table 1 and Figure 1 are recommended to be in italic.

Figure 2 is incomplete in the manuscript. Please provide the full figure of Figure 2, and the figure and figure caption are recommended to be on the same page.

Please use not only the different colors but also the different styles for the legend lines in Figure 3. This comment is also for Figure 8, Figure 9, and Figure 11.

Figure 4 and the caption of Figure 4 should be on the same page.

Figure captions of multiple figures in the manuscript are not readable (e.g. Figure 6, Figure 9, Figure 10). Please update them.

Figure 7 is incomplete. Please provide the complete figure.

Figure 11 is overlapping with its figure caption. Please update the figure.

The authors should explain the diversity gain (DG), envelope correlation coefficient ECC, and mean effective gain (MEG) with proper reference to show the robustness of the proposed MIMO antenna system. As these parameters are not explained properly. Here are some suggested articles for the authors. Isolation and Gain Improvement of a Rectangular Notch UWB-MIMO Antenna, Sensors, 2022; and Isolation enhancement of a metasurface-based MIMO antenna using slots and shorting pins, IEEE Access, 2021. What are the ideal values for the MEG, DG, and ECC of a MIMO antenna? Add them to the manuscript.

As the isolation of the MIMO antenna is significantly affected by the antenna element spacing. Please add the minimum element spacing column in the comparison table (Table 2) for a fair comparison. And, please add more recently published 4-port MIMO antenna in the comparison table

The references are not formatted accordingly to the journal template. The Journal name should be in italics instead of the article’s title. And, the year should be bolded not the volume number.

Round 2

Reviewer 2 Report

Some of the comments of the first review have not been answered properly, which are described below, according to the Response to Reviewer-2.

1- (Point 1): "T-shaped" have not been replaced with "L-shaped" in the abstract (Line 2).

2- (Point 4): your response is : To further clarify this point, Figure 10 has been included in the manuscript. It can be observed from Figure 10 that the currents almost do not move between the two antenna elements for the “STEP 3” as compared to the surface current distribution in “STEP 2” in Figure 9.  

what is the reason?

3- (Point 7): What has changed in Figure 1? I am still confused about slot "C" and slot "E".

4- (Point 9): I think the authors didn't understand what we meant. Show the X and Y axes in figure 1.

Reviewer 3 Report

Some updates are required before final submission.

1.  Figure 7 is too big. Please reduce the size of the Figure.

2. In Figure 12 for ECC and Diversity gain the x-axis is not visible. Updates are required for this figure. Please provide a clear graph.

3. Some responses are not included in the updated manuscript (Response 3 & Response 13). In the response letter, the authors responded that they updated accordingly in the manuscript. However, those are not added in the revised manuscript. As the isolation of the MIMO antenna is significantly affected by the antenna element spacing, add the minimum element spacing column in the comparison table (Table 2) for a fair comparison.
